# Priority U-Net: Detection of Punctuate White Matter Lesions in Preterm Neonate in 3D Cranial Ultrasonography

**Pierre Erbacher**[1]                                         PIERRE.ERBACHER@CREATIS.INSA-LYON.FR
[1] *Univ Lyon, INSA-Lyon, Université Claude Bernard Lyon 1, UJM-Saint Etienne, CNRS, Inserm, CREATIS UMR5220, U1206, F69621 LYON, France*

**Carole Lartizien**[1]                                       CAROLE.LARTIZIEN@CREATIS.INSA-LYON.FR

**Matthieu Martin**[1]                                        MATTHIEU.MARTIN@CREATIS.INSA-LYON.FR

**Pedro Foletto Pimenta**[1]                                  PEDRO.PIMENTA@CREATIS.INSA-LYON.FR

**Philippe Quetin**                                           PQUETIN@CH-AVIGNON.FR
[2] *CH Avignon, France*

**Philippe Delachartre**[1]                                   PHILIPPE.DELACHARTRE@CREATIS.INSA-LYON.FR

**Editors:** Under Review for MIDL 2020

## Abstract

About 18-35% of the preterm infants suffer from punctuate white matter lesion (PWML). Accurately assessing the volume and localisation of these lesions at the early postnatal phase can help paediatricians adapting the therapeutic strategy and potentially reduce severe sequelae. MRI is the gold standard neuroimaging tool to assess minimal to severe WM lesions, but it is only rarely performed for cost and accessibility reasons. Cranial ultrasonography (cUS) is a routinely used tool, however, the visual detection of PWM lesions is challenging and time consuming due to speckle noise and low contrast image. In this paper we perform semantic detection and segmentation of PWML on 3D cranial ultrasonography. We introduce a novel deep architecture, called Priority U-Net, based on the 2D U-Net backbone combined with the self balancing focal loss and a soft attention model focusing on the PWML localisation. The proposed attention mask is a 3D probabilistic map derived from spatial prior knowledge of PWML localisation computed from our dataset. We compare the performance of the priority U-Net with the U-Net baseline based on a dataset including 21 exams of preterm neonates (131 PWMLs). We also evaluate the impact of the self-balancing focal loss (SBFL) on the performance. Compared to the U-Net, the priority U-Net with SBFL increases the recall and the precision in the detection task from 0.4404 to 0.5370 and from 0.3217 to 0.5043, respectively. The Dice metric is also increased from 0.3040 to 0.3839 in the segmentation task.

**Keywords:** Soft attention, U-Net, Detection, 3D Ultrasound, Preterm Neonates

## 1. Introduction

Brain damages, particularly of cerebral white matter (WM), observed in premature infants in the neonatal period are responsible for neurodevelopmental sequelae in early childhood (Pierrat et al., 2017). Punctuate white matter lesions (PWML) are the most frequent WM abnormalities, occurring in 18–35% of all preterm infants (Nguyen et al., 2019) (Tusor et al., 2017). Accurately assessing the volume and location of these lesions during the early postnatal period would help paediatricians adapting the therapeutic strategy which aims to limit the occurence of neurodevelopmental disorders. MRI is the gold standard neuroimaging modality to detect minimal to severe WM lesions, but it is rarely performed for cost and accessibility reasons. On the contrary, cranial ultrasonography (cUS) is routinely used, however, the visual detection of PWM lesions is challenging and time consuming because these lesions are small (in our dataset, the median volume of the lesions is 4 mm$^3$) with variable contrast and have no specific pattern. In addition, lesion location is difficult to determine because of the important variability of the brain anatomy at this age.

Research on automatic detection of PWML in MR images was initiated by Mukherjee (Mukherjee et al., 2019) using standard image analysis methods. One other team has recently tackled this issue based on a deep architecture (Liu et al., 2019b). Despite the high contrast and low noise of MR images, the reported accuracy for the PWML detection task remains low with a Dice under 0.60 and a recall at 0.65 for the best published model. As far as we know, there is currently no known research team working on automatic segmentation of PWML on cUS data. This task is very challenging. Indeed, US images are difficult to analyse because of their low contrast, the presence of speckle and the high variability related to the data acquisition process.

In this paper, we introduce a novel deep architecture based on the U-Net (Ronneberger et al., 2015) backbone to perform the detection and segmentation of PWMLs in cUS images. This architecture combines a soft attention model focusing on the PWMLs location and the self balancing focal loss introduced by Liu (Liu et al., 2019a). The soft attention mask is a 3D probabilistic map derived from a spatial prior knowledge of PWMLs location computed from our dataset. The article is structured as follows. In a first part, we describe our dataset, then we introduce the Priority U-Net taking spatial prior knowledge of PMWLs as input and we compare it with the U-Net using appropriate metrics and visualization of 3D reconstruction of predicted PMWLs.

## 2. Method

### 2.1. Data description

In this study, we used 21 3D reconstructed US brain volumes of preterm babies whose mean age at birth was $31.6 \pm 2.5$ gestational weeks. These volumes were reconstructed from 2D freehand cUS acquisitions using the reconstruction algorithm proposed by Martin et al (Martin et al., 2018). The acquisitions were performed by the paediatrician through the anterior fontanel with an Acuson Siemens 4-9 MHZ multi-D matrix transducer in a coronal orientation with rotation from the front to the posterior of the crane and with a constant velocity.

All the volumes were first centered on the corpus callosum splenium, then cropped from the center to obtain the same size of 360x400x380 voxels with an isotropic spatial resolution of 0.15 mm. A cUS (figure 1a) and a MR axial image (figure 1b) corresponding to the same patient and containing PWMLs are given to highlight the specificity of cUS images compared to MRI. The cUS image has better spatial resolution but has speckle noise and shows many microstructures equivalent to lesions in size and intensity.

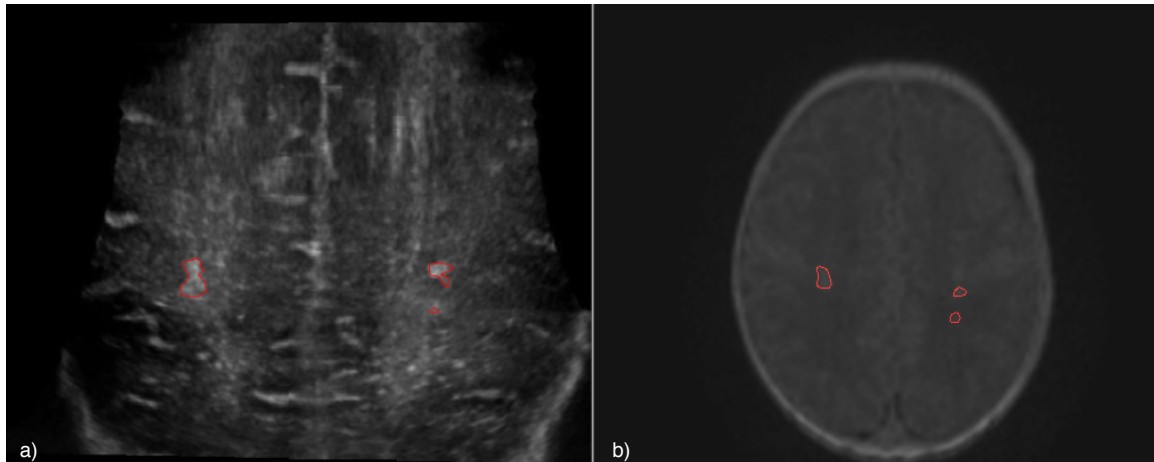

Figure 1: Axial slices extracted from the cUS a) and MR images b) of the same patient. Segmented PMWL are highligted in red. The cUS image was segmented by an expert pediatrician and the MR image was automatically segmented with the algorithm proposed by Liu (Liu et al., 2019b).

### 2.2. Lesion description

The PWMLs were manually delineated on the cUS images by an expert pediatrician. The lesions were visible in approximately 3000 coronal images, an example of such image is shown in figure 2.2. After their manual segmentation, the lesions were isolated by identifying the connected components within each volume (26 connectivity), which resulted in the creation of 547 clusters. Among these, we retained the clusters with a volume bigger than 1.7 mm$^3$, which defined the 131 PWMLs that compose our database. Their volume range from 1.75 mm$^3$ to 61.09 mm$^3$ with a median size of 4 mm$^3$. As shown in figure 3a and figure 3b, most of the lesions remain small : 25 % of the largest PWMLs represent 62.5 % of the PWML total volume.

Bivariate density estimations of the projection of the PWMLs on the axial, sagital and coronal planes were computed. The corresponding density maps are respectively shown in figures 4a, 4b and 4c illustrating that PWMLs are preferentially located around the ventricular systems as previously reported by (Guo et al., 2017).

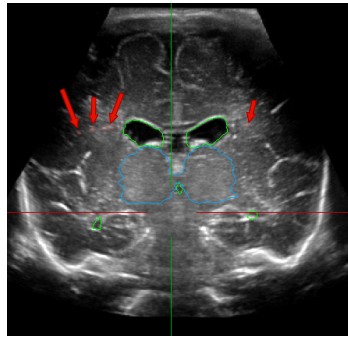

Figure 2: Example PWML (red), thalamus (blue) and ventricular system (green) visible in a coronal slice

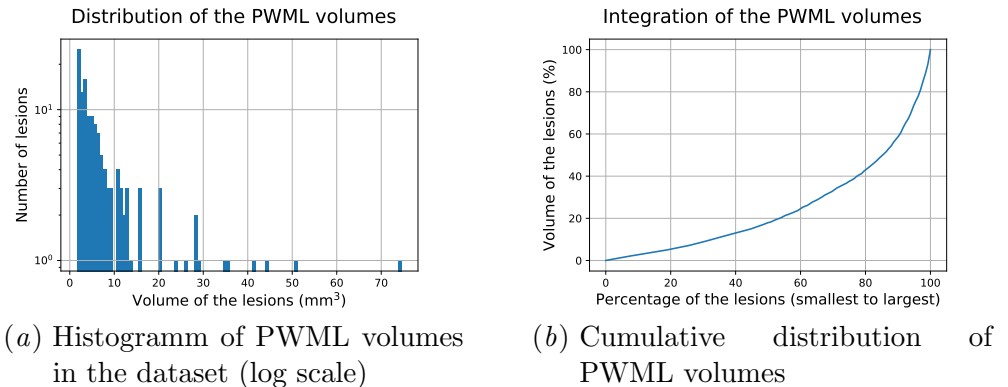

($a$) Histogramm of PWML volumes in the dataset (log scale)

($b$) Cumulative distribution of PWML volumes

Figure 3: PWMLs volume analysis: Most of the PWML are small but only represent a small fraction of the volume.

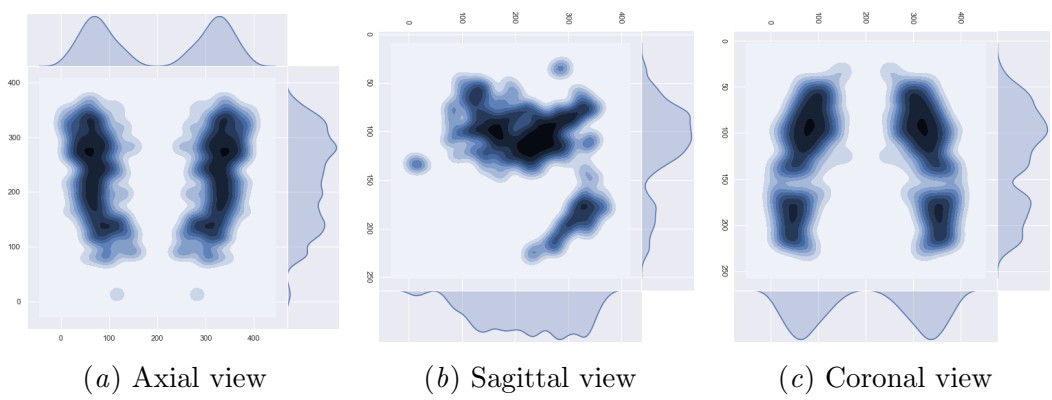

($a$) Axial view      ($b$) Sagittal view      ($c$) Coronal view

Figure 4: Multivariate density estimation of axial, sagittal and coronal projections of PWMLs.

## 2.3. Priority U-Net

Our main goal is to use our prior knowledge about the PWMLs location, as illustrated in figures 4, to enable the U-Net to focus on the brain regions with high PWMLs density values.

### 2.3.1. NETWORK DESCRIPTION

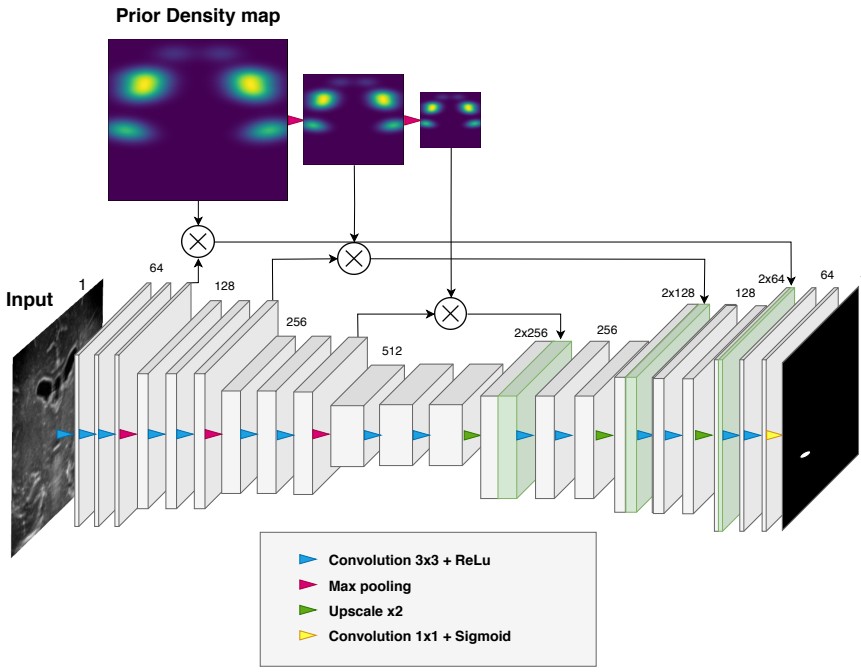

Figure 5: Priority U-Net: The backbone architecture is a 2D U-Net. The proposed multiplicative attention gates are an element-wise multiplication between the skip connection feature-maps and the normalized prior density map.

Our model depicted in figure 5 is based on the attention U-Net proposed by (Oktay et al., 2018). Instead of using self-trained attention gate, we introduce a prior PWMLs density map computed on our training data as detailed in the next section. These maps are fed trough gates at different levels in the U-Net. The gate is an element-wise multiplication, within the skip connections, of the feature map from the U-Net encoding branch by the prior density map.

Our loss term is based on the combination of the three following functions : the binary cross-entropy (1), the Dice (2) and the self balancing focal loss (3) recently introduced by (Liu et al., 2019a) to address learning with highly unbalanced classes:

$$BCE(p, \hat{p}) = -(plog(\hat{p}) + (1 - p)log(1 - \hat{p})),$$ (1)

$$Dice(p, \hat{p}) = 1 - \frac{2p\hat{p}}{p + \hat{p}},$$ (2)

$$SBFL(p, \hat{p}) = \beta \times SBFL_1 + (1 - \beta) \times SBFL_0, \tag{3}$$

with

$$\beta = \frac{0.4 \times \sum(SBFL_0)}{\sum(SBFL_0) + \sum(SBFL_1)} + 0.5$$

$$SBFL_0(p, \hat{p}) = -\hat{p}^\gamma \times (1 - p)log(1 - \hat{p} + \epsilon)$$

$$SBFL_1(p, \hat{p}) = -(1 - \hat{p})^\gamma \times plog(\hat{p} + \epsilon)$$

where $\hat{p}$ and $p$ are respectively the output probability map of the model and the ground truth normalized image. The fixed parameter $\gamma$ is introduced in the focal loss (Lin et al., 2017) to decrease the computed loss on well classified examples, ie with predicted probability close to 1, and increase it on hard examples. $\gamma$ is set to 1 in our experiment. $\epsilon$ is a small constant preventing large loss value. $\beta$ is a parameter balancing the contribution of the positive ($SBFL_1$ corresponding to the lesion) and negative ($SBFL_0$ corresponding to the background) loss terms. Unlike the original focal loss introduced by lin et al, this parameter value changes as a function of the positive and negative loss terms during training. The constants 0.4 and 0.5 allow constraining $\beta$ such that $\beta \in [0.5, 0.9]$. $\gamma$ is a power term applied to the predicted probability that reduces the loss contribution for 'easy' example, ie with predicted probability close to 1, thus increasing the importance of correcting misclassified examples.

In this study, we considered two configurations for the loss of the U-Net and priority U-Net, the first one is the sum of the BCE and Dice losses, the second one is the sum of BCE and SBFL losses.

### 2.3.2. Estimation of the PWMLs density map

The PWML density maps are computed from the concatenation of all training patients volumes. Instead of computing a 3D PWML probability density map (PDM) on the whole volume, we divided the volume into Q batches of N consecutive coronal slices. For each batch, we computed the associated 2D PDM. As a result, we extracted a total of Q 2D PDM for the volume. The bivariate Kernel estimation was computed using the Parzen-Rosenblatt estimator. Let $C_k$ the set of pixel coordinates labeled as PWML for the coronal slice k:

$$C_k = \{(h_0, w_0), ..., (h_p, w_p)\}.$$

Let N the number of consecutive slices in a batch (N = 20 in our experiment) and $i \in [1, Q]$ the $i_{th}$ batch, each batch of slices as defined as $B_i = \bigcup_{k=N \times (i-1)}^{N \times i - 1} C_k$. Because N is small, we consider that the set of points $B_i$ of size $M_i$ are coming from the same distribution. The PWML density $\hat{P}_i(\mathbf{z})$ at point $\mathbf{z}$ for set of point $B_i$ of size $M_i$ is then computed as

$$\hat{P}_i(\mathbf{z}) = \frac{1}{M_i} \sum_{j=1}^{M_i} k_h(\mathbf{x}_j^i - \mathbf{z}), \tag{4}$$

where $k_h$ is a centered Gaussian kernel of fixed width $h$ and $\mathbf{x}_j^i = (h_j^i, w_j^i)$ is the coordinate of point $j$ in the set $B_i$.

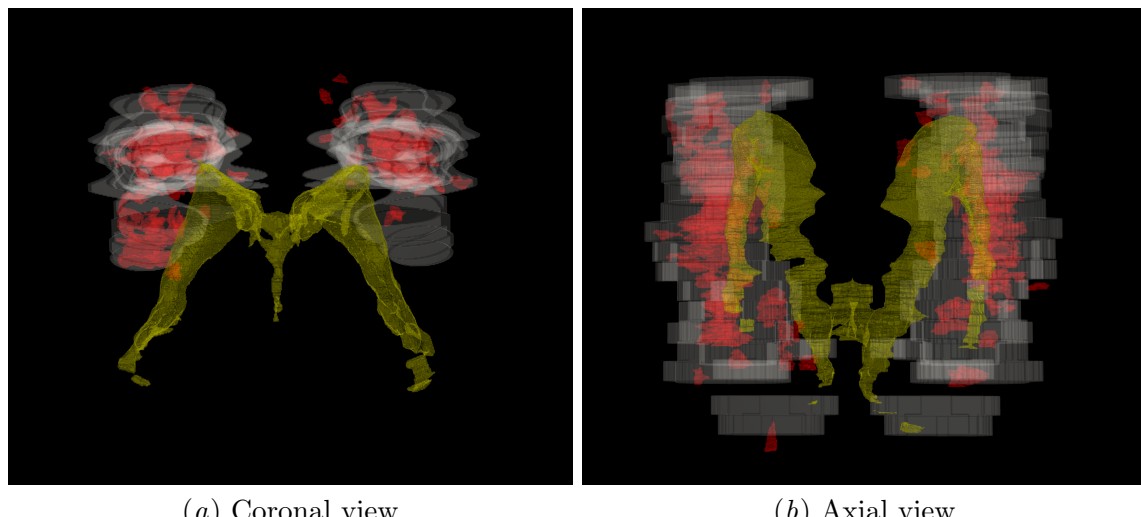

(*a*) Coronal view          (*b*) Axial view

Figure 6: Axial view of 3D PWML (red) superposition. Density map thresholded for visualization (white), ventricular system (yellow).

### 2.4. Experiments

We performed an ablation study in order to evaluate the impact of the attention module and of the self balancing focal loss on the performance of Priority U-Net, thus leading to the evaluation of four models : U-Net with standard binary cross-entropy and Dice loss (U-Net(BCE+dice)), U-Net with self balanced focal and Dice losses (U-Net(SBFL+dice)), Priority U-Net with standard binary cross-entropy and Dice losses( Priority U-Net(BCE+dice)), Priority U-Net with self balanced focal and Dice losses (Priority U-Net(SBFL+dice)).

#### 2.4.1. IMPLEMENTATION DETAILS

The entire pipeline was implemented in python with Tensorflow/ keras libraries. For each model, we performed a 10-fold cross validation with 2 patients in the validation set and 19 patients in the training set. All networks were trained on the 360x400 cropped images. The initial learning rate was fixed at $10^{-4}$ with the Adam optimizer.

#### 2.4.2. PERFORMANCE EVALUATION

Detection performance was evaluated at the lesion level by deriving 3D lesion maps from the labeled maps outputted by Priority U-Net. A 26-connectivity rule was used to identify the connected components. As for the training data, detected lesions smaller than $1.7mm^3$ (600 voxels) were removed. A lesion was considered as a true positive if it intersected a true PWML by at least one voxel. Otherwise, it was considered as a false positive. Detection performance were reported in terms of Precision and Recall.

We also evaluated the segmentation performance by estimating the precision $P_V^i$ defined as the ratio of the predicted lesional volume for patient $i$ over the true lesional volume and

the recall $R_V^i$ defined as the ratio of the predicted lesional volume over the detected lesional volume.

For each patient $i$, we then computed the scalar $\alpha_i$ representing the fraction of true lesional volume for this patient over the total lesional volume in the database. This allowed computing the mean of the precision, recall and dice weighted by the fraction of lesion volume of each patient as follows:

$P_V = \sum_{i=1}^N \alpha_i P_V^i,$
$R_V = \sum_{i=1}^N \alpha_i R_V^i.$

We also report 3D Sørensen-Dice values, although this pure segmentation metric does not fit our objective.

## 3. Results

Table 1: Lesion detection performance.

| Model | Precision | Recall |
|---|---|---|
| U-Net (BCE + Dice) | 0.4404 | 0.3217 |
| U-Net (SBFL + Dice) | 0.2347 | **0.5510** |
| Priority U-Net (BCE + Dice) | 0.4464 | 0.4347 |
| Priority U-Net (SBFL + Dice) | **0.5370** | 0.5043 |

Table 2: Lesional volume estimation (Segmentation).

| Model | Precision | Recall | Dice | Specificity |
|---|---|---|---|---|
| U-Net (BCE + Dice) | 0.5004 | 0.2419 | 0.3040 | 0.9999 |
| U-Net (SBFL + Dice) | **0.6043** | 0.1806 | 0.2611 | 0.9999 |
| Priority U-Net (BCE + Dice) | 0.5455 | 0.2789 | 0.3565 | 0.9999 |
| Priority U-Net (SBFL + Dice) | 0.5289 | **0.3206** | **0.3839** | 0.9999 |

Detection performance reported in Table 1 indicate that Priority U-Net combined with the self balancing focal loss achieves the best precision of 53.7%. It outperforms the baseline U-Net both in terms of precision (0.5370 versus 0.4404) and recall (0.5043 versus 0.3217). The positive impact of the self balancing loss on the performance of Priority U-Net is also underlined by a significant increase of both precision (0.5370 versus 0.4464) and recall (0.5043 versus 0.4347). Effect of SBFL on the U-Net model also positively impacts the recall but significantly degrades the precision respectively (0.3217 vs 0.5510) and (0.4404 vs 0.2347). Note that Priority U-Net with SBFL produces a few more false positives detection than the U-Net with SBFL (recall of 0.5043 versus 0.5510). This however, is largely counterbalanced by the significant increase in precision.

Regarding the estimation of the lesional volume reported in Table 2, Priority U-Net (dice+SBFL) compared to the baseline U-Net increases both precision (0.5289 vs 0.5004) and recall (0.3206 vs 0.2419).

The prior density gate has positive impact both on the detection and segmentation performance. Conclusion regarding the impact of the self balancing focal loss is less clear-cut. For Priority U-Net, SBFL has a positive impact on both detection and segmentation

metrics. For the U-Net architecture, SBFL increases the precision but degrades the recall for both the detection and PWMLs volume estimation.

Figure 7: Axial view of a patient with good recall and precision

Figure 7 illustrates the visual performance achieved by the four evaluated models for the detection of PWMLs in one patient.

## 4. Discussion - Conclusion

The detection of PMWL in cUS is challenging due to the high class imbalance and the low contrast. The high variance in both PWMLs size, echogenicity and speckle noise makes the detection task difficult, especially to differentiate PWMLs from blood vessels. Priority U-Net achieved fairly good detection performance, with a recall and precision of 50.43% and 53.70%, respectively. In 2019, Mukherjee (Mukherjee et al., 2019) initiated the PWMLs detection on preterm infants on MRI. Depending on the recall/precision trade-off, their model achieved a recall from 6.92% to 49.77% and a precision from 7.32% to 52.86%. As far as we know, we are the first to show that we can achieve similar performance in 3D cUS.

The soft attention gate of Priority U-Net boosts detection where the PWML density map has high value. On the contrary, area with very high echogenicity are not considered if the lesion density map has low probability values in this region. Therefore it reduces the number of false positive detections, thus increasing precision. However, as observed in figure 7, this may also induce recurrent false positive detections at the back of the ventricular system where PWML density is high. Some false positive detections also occurred close to the ventricular system borders because of high intensity voxels in this area.

Results reported in lines 1 and 3 of Table 1 compare lesion detection performance achieved by a U-Net architecture associated to standard loss (cross entropy and dice loss), respectively without (line 1) and with (line 3) the addition of the proposed attention map. This comparison underlines the impact of the proposed attention map. We report a gain in sensitivity (recall) from 0.32 to 0.43 while keeping a constant precision of 0.44. This result shows the benefit of the proposed attention map on sensitivity. Results reported in lines 1 and 2 of Table 1 compare the influence of the self-balanced focal loss (SBFL) on the performance of the standard U-Net architecture (ie without attention map). This shows that SBFL allows a significant gain in sensitivity (recall from 0.32 to 0.55) at the price of a drop in specificity (precision from 0.44 to 0.23). The comparative analysis of lines 2 and 4 of Table 1 is more complex since these architectures combine the influence of two parameters: the attention maps and the self-balanced focal loss. Combining the two previous analyses, we may conclude that the attention map controls the drop in specificity induced by SBFL while preserving sensitivity. Also note that SBFL has been introduced very recently by Liu et al (Liu et al., 2019a). The sensitivity gain observed on the task of PWML lesion detection in MRI motivated our choice to consider this new loss term. However, we use the same hyperparameters values as in (Liu et al., 2019a) which may not be optimal for our application.

We tried other strategies to incorporate the prior attention maps. First, we fed them as a second channel in the input image. The result were equivalent to that achieved with U-Net presumably because the prior map was degraded with successive non-linear transformation through the network. We also tried to add them on the last decoding stage, again there was no improvement with regards to U-Net.

Regarding segmentation, performance achieved with Priority U-Net on cUS images are far from that achieved with the best published model in MRI (Liu et al., 2019b). On the Dice score, their model indeed performed 21.47% better. This was expected, as we are working on much noisier images. Also note that a direct comparison is not straightforward. These difference may be due to the different voxel size resolution of the two modalities, less than 0.04 $mm^3$ for cUS versus around 0.8 $mm^3$ for MRI (Nguyen et al., 2019). This may also explain the difference in the median lesion size observed of 30 $mm^3$ with MRI (Tusor et al., 2017) compared to 4 $mm^3$ in our study based on cUS. We would like to emphasize that we are at the early stage of this study, so that we cannot estimate how the accuracy of lesional volume may impact the clinical follow-up.

Perspective include analysing the impact of the SBFL hyperparameters on detection performance with standard U-Net and Priority U-Net as well as constraining the model with a more appropriate loss function to penalize predictions too close to the ventricular system. We also would like to compare the map learned by the attention gate from Oktay (Oktay et al., 2018) with our prior density maps. Finally, we plan to increase the size of

our dataset and design a middle-term cross validation study with MRI to gain insight on the accuracy of the lesional volume we can achieve with both modalities.

## Acknowledgments

This work was supported by the LABEX PRIMES (ANR-11-LABX-0063) and performed within the framework of the LABEX CELYA (ANR-10-LABX-0060) of Université de Lyon, within the program "Investissements d'Avenir" (ANR-11-IDEX-0007) operated by the French National Research Agency (ANR).

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
