# OpenReview forum: "Priority Unet: Detection of Punctuate White Matter Lesions in Preterm Neonate in 3D Cranial Ultrasonography"
_MIDL.io/2020/Conference — MIDL 2020_

### Official Review · AnonReviewer1 · 2020-03-06
**Detection of large PWM lesions in 3D cranial US**

**Rating:** 3
**Confidence:** 3
**Recommendation:** Oral

**Summary:**

Authors present their work on detecting large PWM lesion in 3D cranial US images. They have implemented a U-net for this and introduce prior density maps into this network, to improve overall detection performance. Experiments demonstrate the original vs modified U-net with different loss functions. The best method reaches a sensitivity of 50% with good precision.

**Strengths:**

This appears to be the first work on PWM lesion detection in 3D cranial ultrasonography. This application is important and the work contributes well to this problem. The gold standard is MRI, which is not always available and hence cUS is used. The inclusion of prior maps into the u-net is a good addition.

**Weaknesses:**

The main weakness of this work is the selection of lesions that are included in the evaluation and results. While the median lesion size is 0.413 mm3, authors only include lesions larger than 1.700 mm3. Figure 3 is not very clear on this, but this is obviously a small fraction of the lesions. This raises two questions: (1) is the problem still relevant if only large lesions can be found and (2) is an automated method needed to find large lesions, since there are very easy to find visually owing to their large size.

**Justification Of Rating:**

The application is very original and well chosen by the authors. The methodological contribution is not that large, but including priors into a method is a nice idea and this is an ok solution. The performance of the method is moderate, but since this is a very hard task and one of the first methods on this, it deserves to be presented/published and will likely inspire future works that will improve upon this.

**Paper Type:**

validation/application paper

**Questions To Address In The Rebuttal:**

Authors might answer my two questions above: (1) is the problem still relevant if only large lesions can be found and (2) is an automated method needed to find large lesions, since there are very easy to find visually owing to their large size.

What kind of pre-processing is involved? Are images aligned into a standard space? I assume they are, otherwise the prior maps would not be aligned with the data.

How does the methodological contribution (prior maps) relate to other methods proposed? For example methods for adult brain MRI segmentation that involve prior maps?

In section 2.3.2, please explicitly mention again that the prior maps are based only on the training data, not including the test data. Currently it mentions 'all patient volumes'.

What is the performance of human observers on this task? How does this relate to the performance of automatic methods? If human performance is low/moderate, how do authors guarantee a good ground truth?

**Special Issue:**

no

---

> ### Author Response · Authors · 2020-03-27
> **We thank the reviewer for his feedback. Here are some answers to the questions addressed in the rebuttal**
>
> Answer to question 1.  Authors might answer my two questions above ...size
>
> Medical doctors are interested in lesion volume and precise localization. The volume of lesions allows the calculation of the lesion load in MRI (Nguyen et al 2019) and could allow in cranial ultrasound to calculate a biomarker taking into account thalamic volume and lesion volume, inspired by the work of Tusor et al (Tusor et al., 2017).
>
> Nguyen, A.L.A., et al, 2019. The brain’s kryptonite: Overview of punctate white matter lesions in neonates. Int. J. Dev. Neurosci. 77, 77–88.
> Tusor, N., et al, 2017. Punctate White Matter Lesions Associated With Altered Brain Development And Adverse Motor Outcome In Preterm Infants. Sci. Rep. 7, 13250.
> To answer your first question, our objective is to provide the most accurate estimation of the lesion load. We chose to discard the small lesions both from the training and testing phase.
>
> To answer your first question, our objective is to provide the most accurate estimation of the lesion load. The first reason we chose to discard the small lesions both from the training and testing phase is that small lesions may be confused with other small structures, the second is that they contribute less significantly to lesion volume and the third reason is that they may not be seen on MRI due to differences in resolution. So, to compare the 2 modalities one possibility would be to filter them.
>
> As also answered to reviewer 4, we would like to emphasize that we are at the early stage of this study. So that we cannot estimate how the accuracy of this biomarker may impact the clinical follow-up. Moreover, we would like to underline the difference in the median lesion size observed with MRI (Tusor et al., 2017) : 30 mm3 compared to 0.41 mm3 in our study based on cUS. These difference may be due to the different voxel size solution of the two modalities, less than 0.01 mm3 for cUS versus around 0.08 mm3 for MRI (Nguyen et al 2019).
>
> To answer your second question, large lesions may be “easy” to detect in MRI but not so easy in cUS because of misleading signal patterns that may induce false positives. Moreover, whatever their size, the lesion annotation process is quite long (1 hour per 3D volume for our expert paediatrician) so that accurate automated methods are welcome to reduce the clinical workload.
>
> Answer to question 2. What kind of preprocessing.. with the data.
>
>  At this stage of the cerebral development, structures in the brains continuously growing and moving. The structures' growth rate is varying a lot between infants. This result with structures with high volumes and shapes variation between patients, so that a non-rigid coregistration to some atlas is not feasible.  As mentioned in the Data description part, we thus only performed rigid registration by centering all the volumes on the corpus callosum splenium.
>
> Answer to question 3: How does the methodological contribution... involve prior maps?
>
> The network architecture relies on the Attention UNet proposed by Oktay et al, the main difference being that we input prior attention maps while these maps are learned in the paper by Oktay et al. As far we know, there is no similar approach involving prior maps for the task of lesion detection. The literature in the domain of priors maps for segmentation include, as an exemple,  the contribution  by Oktay et al for cardiac segmentation, the study by Ganaye et al (Media 2019) for brain structure segmentation or that by Kervadec et al (Media 2019) on integrating a priori constraints like organ size.
>
> Answer to question 4: In section 2.3.2, please explicitly ..volumes'.
>
> This will be corrected in the revised version of the paper
>
> Answer to question 5: What is the performance of human observers on this task? How does this relate to the performance of automatic methods? If human performance is low/moderate, how do authors guarantee a good ground truth?
>
> For this preliminary study, the contours of the PWML were drawn once by a single pediatrician. We cannot therefore calculate criteria for intra or inter-operator variability. In this domain of research which is in its early stage, as underlined above, the reference imaging modality is MRI. As far as we know, only three papers discuss the segmentation of PWML in MRI, but neither of them provides human observer performance analysis for this segmentation task (Liu et al 2019a) (Liu et al 2019b) (Mukherjee et al., 2019).
>
> Yalong Liu, et al Trident segmentation cnn: A spatiotemporal transformation cnn for punctate white matter lesions
> segmentation in preterm neonates. Arxiv, 10 2019a.
>
> Yalong Liu, et al Refined segmentation r-cnn: A two-stage convolutional neural network for
> punctate white matter lesion segmentation in preterm infants. MICCAI  2019b.
>
> Mukherjee, S., Cheng, I., Miller, S., Guo, J., Chau, V., Basu, A., 2019. A Fast Segmentation-free Fully Automated Approach to White Matter Injury Detection in Preterm Infants. Med. Biol. Eng. Comput. 57, 71–87.

---

### Official Review · AnonReviewer4 · 2020-03-14
**Strong accept: the results are promising, good visualizations, some vague parts in the method which should be addressed in the updated version of the paper**

**Rating:** 4
**Confidence:** 3
**Recommendation:** Oral

**Summary:**

This paper proposes a novel deep network, priority U-Net, based on the UNet architecture to perform semantic detection and segmentation of PWML disease on 3D cranial ultrasound images from 21 preterm babies.
The performance of the proposed method was compared with the U-Net on a dataset consist of 547 images. Recall and precision improved in the detection task and the Dice metric is also increased in the segmentation task.

**Strengths:**

The paper is well-written with appropriate introduction.
The size of dataset ( 21 babies) is relatively good.
Results are promising with superior performance compared to the previous work
good visualization of the method/results

**Weaknesses:**

There are some vague parts which I explained to be addressed in the rebuttal and paper.
There are some arbitrary parameters that I miss the rational behind them. For example, how did the author determine the level which the attention maps should be fed to ...



**Justification Of Rating:**

The paper is well written, the results are promising and better performance was achieved compared to the previous work on this problem. The references are up to date, and the results were visualized with various figures which provides the reader with faster understanding.

**Paper Type:**

both

**Questions To Address In The Rebuttal:**

1- In fig 1, the segmentations are not identical in two images. Is it because of the inaccurate manual segmentation of the expert due to the low resolution of the ultrasound? Is the segmentation of the MR accurate then?

2- Is the input a 3D volume of data? I think fig 5 can be misleading with showing only a 2D slice as the input.

3- How did you decide at what level the attention maps should be fed to the network? I meant how did you determine the location of the gate?

4- using 15 patients for training and 4 for validation



**Special Issue:**

yes

---

> ### Author Response · Authors · 2020-03-27
> **We thank the reviewer for his feedback. Here are some answers to the questions addressed in the rebuttal**
>
> Answer to question #1. As stated in the introduction of our paper, image analysis of PWM lesions in neonates is at its early stage. There are only few recent clinical papers as well as methodological papers assessing the feasibility to segment PWML in MRI. As far as we know, there are very no studies focusing on the use of cUS for this task.
> The objective of figure 1 was to illustrate that we can retrieve similar types information from the MRI and cUS images but also underline, as reported by the authors, that PWM lesions may look different in both modalities. The observed difference may explained by the following reasons:
> -cUS and MRI were acquired on the same patient but not the same day, which may partly explain the difference in size, knowing that the cerebral development is fast.
> -The two modalities have different voxel volume, less than 0.01 mm3 for cUS and 0.08 mm3 for MRI (Nguyen et al 2019), which will thus impact the accuracy of the lesion segmentation.
> -Segmentation of the cUS image was performed manually by one expert paediatrician while MRI was automatically segmented by the method proposed by Liu et al with a mean dice coefficient of 0.5986 which is far from precise.
>
> Answer to question #2. Our proposed network is based on a 2D analysis of the coronal slices extracted from the reconstructed 3D cUS volume. Fig 5 is thus correct. We will reword the revised version of this paper to clarify this point.
>
> Answer to question #3. At first we naively tried to feed these attentions maps as a second channel in the input image. The results were basically the same as for the Unet presumably because the prior map was degraded with successive non-linear transformation. Our objective was indeed to propagate the non-degraded information on lesion-preferred localization up to the decoder. We also tried to add the attention map on the last decoding stage, again there was no improvement on the Unet. The attention Unet proposed by Oktay et al provided an interesting way to satisfy our need by feeding attention maps to the network at the skip connections level.
>
> Answer to question #4. Sorry if we misunderstand your question. Performance evaluation was carried out based on a 10-fold cross validation with 19 patients in the training set and 2 patients in the validation set.

---

### Official Review · AnonReviewer2 · 2020-03-18
**Application of UNets with density maps in segmenting cranial ultrasound with unconvincing results**

**Rating:** 2
**Confidence:** 5

**Summary:**

The authors propose using separately computed density maps as manner of attention in segmenting white matter lesions in cranial ultra sound images. The application of UNet to cranial ultrasound is very interesting. The performance of the proposed method improves in terms of dice, however, the sensitivity is significantly lower compared to a vanilla Unet.




**Strengths:**

Pros:
1. Problem statement is incredibly hard. The results are good considering the difficulty.
2. The paper is an easy read.
3. Density estimation is potentially a good idea. However, the manner of computing is the density maps needs explanation. Are the images registered?
4. The combination of balanced cross entropy, balanced focal loss is less used in the community. The results are encouraging.


**Weaknesses:**

Cons:
1. Comparison to self-attention mechanism is missing since that is the main change in the paper.
2. Severe dip in the sensitivity. This could be due to the density maps multiplied with the feature maps so less likely occuring lesions actually get very low attention. This is probably not ideal.
3. More explanations on parameter choices such as gamma and beta is necessary.

**Justification Of Rating:**

The paper does not propose any novel methodology or empirical insight. However, the application is quite novel. This alone does not warrant an acceptance. The results are also not convincing considering vanilla UNets provide better sensitivity indicating the proposed method is not adequate in picking smaller lesions.

**Paper Type:**

validation/application paper

**Special Issue:**

no

---

> ### Author Response · Authors · 2020-03-27
> **We thank the reviewer for his feedback. Here are some answers to the reported weaknesses**
>
> Weakness # 1. We addressed this point in our answer to reviewer 4. Please also refer to it. We tried other strategies to incorporate the prior attention maps:
> First, we naively fed these attentions maps as a second channel in the input image. The result were basically the same as for the Unet presumably because the prior map was degraded with successive non-linear transformation through the network. We also tried to add the attention map on the last decoding stage, again there was no improvement on the Unet.
> We will clarify this point in the discussion section of the final version of the manuscript. The main idea was not to compare against the self-attention mechanism but to correctly incorporate our prior density map in the network. We did try the Attention Unet at a preliminary stage of the project without any convincing result. We guess this is because the learned filter is doing an average of the true distribution across the volume and not learning accurately the attention considering the position of the slice in the 3D volume.
>
> Weakness # 2. Assuming that the reviewer refers to performance results reported in table 1 regarding the detection performance, we indeed report a sensitivity of 0.5043 for Priority U-Net compared to 0.5510 for the U-Net with the SBFL + Dice loss. This should however be counterbalanced by a significant gain in precision, since our model achieves a 0.537 precision compared to 0.23. We agree that a more extensive analysis should be carried out to describe (localization, size etc..) the PWM lesions that were missed by our model in order to get a better understanding of the impact of the prior attention map.
>
> Weakness #3 As stated in section 2.3.1., and observed in the formulation of the loss terms SFBL0 and SFBL1  for the negative and positive classes respectively, gamma is a power term applied to the predicted probability that reduces the loss contribution for ‘easy’ example, ie with predicted probability close to 1, thus increasing the importance of correcting misclassified examples.
> Beta is a parameter balancing the contribution of the positive (SBFL1 corresponding to the lesion) and negative (SBFL0 corresponding to the background) loss terms. Unlike the original focal loss introduced by lin et al, this parameter value changes as a function of the positive and negative loss terms during training.
> This will be clarified in the final version of the paper.

---

> > ### Comment · AnonReviewer2 · 2020-03-29
> > **Drop in sensitivity not explained.**
> >
> > The drop in sensitivity needs to be properly justified. An increased precision is good when one is really concerned about getting boundaries right -- the increased precision explains the better dice. In other words, larger lesions are probably segmented "better". This can probably be achieved by adjusting the hyper-parameters of a vanilla UNet. It is not clear what the method is adding.
> >
> > It was therefore extremely important to show some interesting benefits of the proposed attention maps, which the paper fails at.

---

> > > ### Author Response · Authors · 2020-03-31
> > > **Please find some arguments attesting benefits of our proposed attention maps**
> > >
> > > Results reported in lines 1 and 3 of Table 1 compare lesion  detection performance achieved by a U-Net architecture associated to standard loss (cross entropy and dice loss), respectively without (line 1) and with (line 3) the addition of the proposed attention map. This comparison underlines the impact of the proposed attention map. We report a gain in sensitivity (recall) from 0.32 to 0.43 while keeping a constant precision of 0.44. This result shows the benefit of the proposed attention map on sensitivity.
> > >
> > > Results reported in lines 1 and 2 of Table 1 compare the influence of the self-balanced loss (SBFL) on the performance of the standard UNet architecture (ie without attention map). This shows that SBFL allows a significant gain in sensitivity (recall from 0.32 to 0.55) at the price of a drop in specificity (precision from 0.44 to 0.23).
> > >
> > > The comparative analysis of lines 2 and 4 of Table 1 is more complex since these architectures combine the influence of two parameters: the attention maps and the self-balanced focal loss. Combining the two previous analyses, we may conclude that the attention map controls the drop in specificity induced by SBFL while preserving sensitivity.
> > >
> > > Also note that SBFL has been introduced very recently by Liu et al. The sensitivity gain observed on the task of PWML lesion detection in MRI motivated our choice to consider this new loss term. However, we use the same hyperparameters values as Liu et al which may not be optimal for our application. We will further analyse the impact of these hyperparameters on detection performance with standard U-Net and Priority U-Net.
> > >
> > > To conclude, our results indicate benefits of the proposed attention map both for the U-Net with standard Dice loss and for Priority U-Net with the newly introduced SBFL loss.

---

### Meta-Review · Area_Chair1 · 2020-04-05
**MetaReview of Paper228 by AreaChair1**

**Rating:** 3
**Recommendation For Accepted Papers:** Poster

**Metareview:**

The paper tackles a very challenging problem and presents good results. The rebuttal appropriately addresses questions of the reviewers and justifies the methodological choices made.

**Paper Type:**

methodological development

**Special Issue:**

no

---

### Decision · Program_Chairs · 2020-04-11

Accept